# Hepatotoxicity in Carp (*Carassius auratus*) Exposed to Perfluorooctane Sulfonate (PFOS): Integrative Histopathology and Transcriptomics Analysis

**DOI:** 10.3390/ani15040610

**Published:** 2025-02-19

**Authors:** Lin Tang, Guijie Hao, Dongren Zhou, Yunpeng Fan, Zihao Wei, Dongsheng Li, Yafang Shen, Haoyu Fang, Feng Lin, Meirong Zhao, Haiqi Zhang

**Affiliations:** 1Key Laboratory of Microbial Technology for Industrial Pollution Control of Zhejiang Province, College of Environment, Zhejiang University of Technology, Hangzhou 310014, China; 221122270111@zjut.edu.cn (L.T.); 2112127020@zjut.edu.cn (Z.W.); 2112127175@zjut.edu.cn (D.L.); 201806022103@zjut.edu.cn (H.F.); zhaomr@zjut.edu.cn (M.Z.); 2Key Laboratory of Freshwater Fisheries Healthy Aquaculture, Ministry of Agriculture and Rural Affairs, Key Laboratory of Fish Health and Nutrition of Zhejiang Province, Key Laboratory of Fishery Environment and Aquatic Product Quality and Safety of Huzhou City, Zhejiang Institute of Freshwater Fisheries, Huzhou 313001, China; haoguijie79@126.com (G.H.); zhoudongren82@126.com (D.Z.); yunpengfan1994@163.com (Y.F.); shengyafang1994@126.com (Y.S.); wwlinfeng@163.com (F.L.)

**Keywords:** PFOS, crucian carp, histopathology, lipid metabolism, omics analysis

## Abstract

This study explores the impact of long-term exposure to perfluorooctane sulfonate (PFOS), a pollutant commonly found in water due to its use in many industrial and consumer products. PFOS is concerning because it does not easily break down and can accumulate in living organisms, potentially causing harm. Our research aims to understand how PFOS affects the liver health of crucian carp, which play a vital role in aquatic ecosystems. By exposing juvenile crucian carp to different levels of PFOS for 21 days, we observed significant liver damage, including vein congestion, cell breakdown, and the formation of empty spaces within cells. Transcriptomic analysis revealed that PFOS impacts genes involved in lipid processing, energy production, infection response, and hormone regulation. These changes suggest that PFOS disrupts multiple body systems in fish, leading to conditions similar to fatty liver disease in humans. Our findings underscore the critical need for stricter pollution controls to protect aquatic environments and human health, thereby maintaining ecological balance and ensuring safer food supplies. Moreover, understanding the health impacts of PFOS on fish can inform conservation efforts and policy-making aimed at preserving aquatic biodiversity.

## 1. Introduction

Perfluorooctane sulfonate (PFOS) is a persistent organic pollutant (POP) that poses significant ecological and health risks due to its environmental persistence, long-range transport capacity, bioaccumulation potential, and inherent biological toxicity. Despite not being classified as an endocrine-disrupting chemical (EDC) by the European commission or other regulatory agencies, PFOS has been commonly reported in surface water [1,2,3], wastewater [4,5], and humans, with concentrations in highly exposed groups reaching up to mg/L levels [6,7]. This has triggered a surge in research focusing on the ecological and environmental impacts of PFOS pollution [8,9,10].

In humans, PFOS exposure has been linked to various adverse health effects, including liver damage, thyroid dysfunction, decreased fertility, developmental issues in children, and increased cholesterol levels [11,12,13,14]. Chronic exposure to PFOS has also been associated with potential carcinogenic effects [15]. In wildlife, PFOS exposure can result in severe toxic effects, including organ damage, reproductive dysfunction, and metabolic disturbances [16,17,18]. Fish at the top of the food chain in aquatic ecosystems are particularly susceptible to PFOS accumulation, which can result in substantial alterations in liver-specific protein profiles, increased immune cell populations, and structural damage, eventually triggering fibrosis and compromising immune functions [19].

Although PFOS is generally detected in the environment at very low concentrations, typically ranging from 0.2 to 100 ng/L depending on the location and media [20], its bioaccumulative and biomagnifying properties allow it to concentrate in the bodies of aquatic organisms through respiration and ingestion. The liver, a pivotal organ for metabolism and detoxification, serves as the primary depot for PFOS accumulation. Recent studies have demonstrated pronounced hepatotoxic effects of PFOS in mammals [21,22,23] and fish [19], impacting various biological functions. Chronic PFOS exposure in zebrafish disrupts lipid biosynthesis, fatty acid β-oxidation, and *VLDL/LDL* excretion, culminating in hepatic steatosis [24]. Additionally, chronic exposure has been shown to shrink the liver and affect energy metabolism in carp [25]. However, the fundamental knowledge of PFOS hepatotoxicity in other freshwater fish species and the precise mechanisms of PFOS-induced liver injury remains elusive.

To better understand the molecular mechanisms underlying PFOS-induced hepatotoxicity, we focused on four key categories of genes: fat-related genes, energy metabolism genes, infection response genes, and hormone regulation genes. Fat-related genes, such as those involved in lipid biosynthesis and fatty acid β-oxidation, are crucial for maintaining cellular energy homeostasis and membrane integrity [26]. Energy metabolism genes play a vital role in ATP production and mitochondrial function, which are essential for cellular activities [27]. Infection response genes, including those associated with inflammation and immune responses, are important for defending against pathogens and repairing tissue damage [28]. Hormone regulation genes control endocrine signaling pathways, influencing growth, development, and stress responses [29].

Due to differences in biological properties among species, the varying safety concentration thresholds for PFOS complicate ecological risk assessments. The crucian carp, an economically significant and geographically widespread indigenous species across China, represents an appropriate model organism for evaluating PFOS toxicity and ecological risks due to its broad adaptability [30]. This study aimed to elucidate the toxicological impacts of PFOS on crucian carp through acute and chronic toxicity assays using juvenile specimens. Histopathological examinations assessed microstructural changes in liver tissues induced by PFOS, complemented by transcriptomic sequencing to unravel molecular toxicity mechanisms. Our findings contribute significantly to monitoring potential health risks to aquatic organisms and inform ecological risk assessments, highlighting the importance of protecting aquatic environments and ensuring safer food supplies.

## 2. Materials and Methods

### 2.1. Chemicals and Animals

PFOS (purity ≥ 90%, CAS: 2795-39-3) was sourced from lullaby chemicals (Wuhan, China). A total of 200 juvenile crucian carp (approximately two months old, initial body length of 7.23 ± 0.89 cm, weight of 6.17 ± 1.03 g) were acquired from a specialized aquaculture farm located in Huzhou, Zhejiang Province, China. Prior to the initiation of experiments, these juveniles were placed in a static breeding tank with a volume of 100 L and underwent a two-week adaptation phase, with 15 fish placed in each tank. Throughout this period, critical water parameters were maintained within optimal ranges: dissolved oxygen (DO) ≥ 5 mg/L, a pH of 7.0 ± 0.5, a controlled temperature of 25 ± 1 °C, and a photoperiod set to 14 h of light followed by 10 h of darkness. In addition, fish are fed commercial feed equivalent to 6.0% of their body weight every day to ensure continuous nutritional support, 50% of the water was replaced every day, and dead fish and residual bait were removed in time to maintain good water quality [31]. All experimental procedures involving animal welfare, manipulation, experimentation, and euthanasia strictly adhered to the guidelines established by the Chinese Society of Laboratory Animal Science. The study protocol, including the methods for the humane euthanasia of fish at the end of the experiment, was approved by the Animal Research and Ethics Committee of Zhejiang Institute of Freshwater Fisheries. Euthanasia was performed using an overdose of tricaine methanesulfonate (MS-222), following the recommended concentration and exposure time to ensure the rapid and painless termination of life. This approach ensures compliance with international standards for the humane treatment of laboratory animals.

### 2.2. Determination of 96 h 50% Lethal Concentration (LC_50_)

To determine the lethal concentration of PFOS that kills half of the test population within 96 h (LC_50_), we conducted experiments using crucian carp following OECD guidelines [32]. Initially, we performed a pre-test to identify the range of PFOS concentrations causing no mortality to complete mortality within 24 h. Based on these results, we chose exposure groups with PFOS concentrations of 0, 10, 15, 20, 25, 30, and 35 mg/L. Each group contained 15 fish, and we ran three replicates for each concentration. Before starting the experiment, the fish were fasted for 24 h to standardize their physiological conditions. During the 96 h test, we maintained consistent water conditions without feeding or changing the exposure solution [33]. Fish were considered dead if they did not respond to gentle stimulation at the tail. Observations were made every 6 h to record changes in behavior, signs of poisoning, and deaths. Dead fish were removed promptly throughout the experiment.

### 2.3. Chronic Toxicity Assays and Sample Collection

For chronic toxicity tests, we selected three PFOS concentrations (0.1, 0.5, and 1.0 mg/L) based on the LC_50_ value from the 96 h test and reported environmental levels [34,35,36]. These concentrations, along with a control group without PFOS, were tested in triplicate with 15 fish per replicate. To keep PFOS levels steady, we refreshed the solution every 48 h [37]. After exposing the fish for 21 days, we randomly sampled them. Prior to sampling, fish underwent a 24 h fasting period. Liver tissues were carefully extracted on ice from all fish. For histological examination, nine liver samples per treatment (three per replicate) were preserved in 4% paraformaldehyde. The remaining liver tissues (six per replicate) were quickly frozen in liquid nitrogen and stored at −80 °C for later transcriptome sequencing analysis.

### 2.4. Histopathological Examination

Liver tissues from crucian carp were fixed in 4% paraformaldehyde solution prepared in phosphate-buffered saline (PBS) with a pH of 7.4 for 24 h at room temperature (approximately 25 °C) to preserve tissue architecture. Following fixation, tissues underwent systematic dehydration through a graded ethanol series, ensuring optimal integrity for subsequent processing. Following dehydration, xylene was utilized for clearing, facilitating paraffin infiltration and embedding. Subsequently, the paraffin-embedded blocks were sectioned into slices of 5–10 μm in thickness using a precision microtome (Leica RM2235, Wetzlar, Germany). Post-sectioning, dewaxing was achieved with xylene, followed by dehydration through a graded alcohol series to prepare for staining. Hematoxylin and eosin (HE) staining was meticulously applied to delineate cellular and nuclear features, followed by secure mounting with neutral gum. After air-drying at room temperature, the prepared slides were subjected to high-resolution imaging under a light microscope (Olympus BX53, Tokyo, Japan).

### 2.5. RNA Sequencing and Differentially Expressed Genes (DEGs) Analysis

Transcriptomic profiling was conducted using liver tissues from fish exposed to experimental conditions. Immediately after collection, samples were preserved in RNAlater solution at −80 °C until RNA extraction. Total RNA was isolated with TRIzol reagent (Invitrogen, Waltham, America) following the manufacturer’s protocol, involving homogenization with a TissueLyser II (Qiagen, Hilden, Germany), phase separation with chloroform, and precipitation with isopropanol. The RNA pellet was subsequently washed, dried, and resuspended in RNase-free water.

RNA quality and quantity were evaluated using a Nanodrop 2000 spectrophotometer (Thermo Fisher Scientific, Waltham, MA, USA) for A260/A280 ratios and agarose gel electrophoresis for rRNA band visualization. RNA integrity was quantified with an Agilent 2100 Bioanalyzer (Agilent Technologies, Beijing, China) using the Agilent RNA 6000 Nano Kit (Catalog No.: 5067-1511, Agilent Technologies, Beijing, China). Only RNA samples with RIN values between 5.0 and 10.0 were selected for further processing (Appendix A). High-quality RNA underwent ribosomal RNA depletion using the Ribo-Zero Magnetic Kit (Illumina, San Diego, CA, USA) and fragmentation prior to cDNA synthesis. First-strand cDNA synthesis was performed with random hexamers and reverse transcriptase, followed by second-strand synthesis using the Illumina TruSeq RNA Sample Prep Kit v2 (Catalog No.: RS-122-2101). The resulting double-stranded cDNA was processed through end repair, A-tailing, adapter ligation, and size selection. PCR amplification enriched the cDNA fragments, preparing them for sequencing.

Sequencing was carried out on the Illumina NovaSeq 6000 platform (Shanghai Meiji Biomedical Technology Company, Shanghai, China), generating paired-end reads of 300 bp. Post-sequencing data processing involved several critical steps to ensure robust and reliable results. Raw reads were first filtered to remove low-quality sequences and adapter contamination using Trim Galore with default parameters. Cleaned reads were then aligned to the reference genome of crucian carp using HISAT2, ensuring the accurate mapping of reads to known gene regions. Transcript reconstruction and quantification were performed using StringTie [38], while gene expression levels were quantified as transcripts per million (TPM) using RSEM [39], providing a normalized measure of expression across samples.

Differential expression analysis was conducted using DESeq2 within the R/Bioconductor framework. DESeq2 applies a negative binomial distribution model to account for biological variability and performs multiple testing corrections using the Benjamani–Hochberg method to control the false discovery rate (FDR). Genes with an FDR < 0.05 and |Log2FC| > 1 were classified as significantly differentially expressed, confirming the reliability of the detected transcriptional changes [40]. Functional annotation and pathway analysis were achieved through GO term categorization and KEGG pathway analysis using the DAVID Bioinformatics Resources and KOBAS, revealing gene function networks and pathways affected by PFOS exposure.

### 2.6. Validation of Selected Genes Using Quantitative Real-Time PCR (qRT-PCR)

To authenticate the transcriptomic sequencing data, qRT-PCR was employed to assess the expression levels of 11 genes demonstrating significant differential transcriptional responses. The stability of β-actin, a housekeeping gene, served as an internal control for normalization purposes. Primer pairs for the qRT-PCR assays were carefully designed utilizing the LightCycler96 SW 1.1 software, with annealing temperatures optimized around 60 °C to guarantee specificity and target amplicons sized at approximately 200 bp. The primers were listed in Appendix A. Each sample was individually assayed and replicated thrice to ensure reproducibility and statistical reliability.

### 2.7. Statistical Analysis

Statistical analyses were executed using SPSS 26.0 software (IBM). Data are presented as means ± standard deviation of the mean (SEM) to succinctly convey the central tendency and variability of the experimental outcomes. A probabilistic analysis was performed on the 96 h mortality data of crucian carp, employing the Probit model within IBM SPSS for parameter estimation. This method facilitated the determination of the LC_50_ alongside its 95% confidence intervals (CI), using chi-square tests to ensure the model’s adequacy. The significance of differences among groups was assessed by One-way analysis of variance (ANOVA), with statistical significance set at *p* ≤ 0.05. Post hoc comparisons were made using Tukey’s Honestly Significant Difference (HSD) test to identify significant pairwise differences between treatment groups.

## 3. Results

### 3.1. Determination of LC_50_

Mortality rates in all populations of crucian carp subjected to PFOS exposure escalated within 24 h, revealing a dose- and time-dependent toxicity pattern characterized by an increase in lethality with higher PFOS concentrations and prolonged exposure periods. Through linear regression, we derived the regression equation, thereby quantifying the relationship between PFOS concentration and fish mortality. The LC_50_ curve is shown in Appendix A. As detailed in Appendix A, the LC_50_ values for crucian carp within a 96 h exposure period to PFOS are delineated. Our findings reveal that the 96 h LC_50_ value is 23.17 mg/L. According to the classification criteria outlined in OECD Test No. 203: Fish, Acute Toxicity Test [32], substances with a 96 h LC_50_ falling between 10 and 100 mg/L are classified as moderately toxic. Hence, based on our findings, PFOS exhibits moderate toxicity towards crucian carp under the given experimental conditions.

### 3.2. Histological Observations

We scrutinized the histological alterations in carp liver tissue 21 days subsequent to PFOS exposure. Notably, liver cells in the control group manifested structural integrity, demonstrating a precise arrangement demarcated by clear intercellular boundaries, nuclei that were intact, and a cytoplasm uniformly dispersed (Figure 1(A1–A3)). Upon examination with 25× magnification, hepatocytes in the treatment cohorts revealed steatosis, evidenced by variable-sized intracellular lipid droplets, notably in the 1 mg/L group where large droplets displaced nuclei to the cellular periphery (indicated by red arrows) (Figure 1(B1,C1,D1)). Enhanced observation at 50× unveiled central vein dilation, hemoglobin cell compression, and deformation in the 0.5 mg/L group (yellow arrows denote these features) (Figure 1(B2,C2,D2)). Across all treated groups, hepatocytes exhibited mild swelling, size incrementation, cytoplasmic staining, thinning, and either a reticular or translucent appearance; vacuolation was noted alongside the loss of normal morphology and a looser cellular array, a trend exacerbated with elevated PFOS concentrations (Figure 1(B2,C2,D2)). Further magnification to 100× disclosed signs of nuclear lysis in some hepatocytes of the treatment groups, characterized by pale staining and displacement alongside nuclear membranes, with the 1 mg/L group exhibiting notable nucleolus dissolution and disappearance (green arrows indicate these instances) (Figure 1(B3,C3,D3)). These histological transformations undeniably illustrate that chronic PFOS exposure inflicts liver impairment in crucian carp, with the severity of hepatic histopathology escalating in parallel with PFOS concentration increments, thereby accentuating the dose-responsive hepatotoxic impact of PFOS on this species.

### 3.3. Transcript Assembly and Gene Functional Annotation

Approximately 85.72 Gb of clean data were generated from the 12 collected samples, with each sample yielding a minimum of 6.39 Gb of clean data and featuring Q_30_ (as the percentage of sequenced bases that have a Phred-scaled quality score of 30 or above, indicating a base call accuracy of at least 99.9% and representing an error rate of no more than 0.1%) bases surpassing 93.94%, indicative of high sequencing quality. Individual clean reads from every sample were compared against a designated reference genome, achieving alignment rates ranging from 85.63% to 91.49%. This analysis collectively detected 56408 expressed single genes and 102,234 transcripts.

In the gene annotation process employing GO, genes were systematically categorized according to their involvement in various metabolic pathways. As illustrated in Figure 2, genes from the contrasts of the 0.1 mg/L PFOS treatment group vs. control comparison (L vs. C), the 0.5 mg/L PFOS treatment group vs. control comparison (M vs. C), and the 1.0 mg/L PFOS treatment group vs. control comparison (H vs. C) were annotated within the GO database, spanning three primary domains: biological process (BP), cellular component (CC), and molecular function (MF). Among the 20 subcategories, the top three most frequently annotated BP terms included “Cellular process”, “Metabolic process”, and “Biological regulation”. For CC, the prevailing categories were “Cell part”, “Membrane part”, and “Organelle”, reflecting the cellular compartments most affected. In the MF domain, “Binding” and “Catalytic activity” emerged as the chief annotations, implicating these functions as pivotal targets and mechanisms responsive to PFOS exposure.

### 3.4. DEGs Analysis

In our study, Pearson correlation analysis was conducted on carp liver samples to assess data consistency (Figure 3A). Notably, sample H_2 displayed a distinctively reduced correlation coefficient compared to its counterparts within the same exposure group. In response to this observation and to ensure consistent quality across all samples, we revisited our experimental procedures and decided to exclude sample H_2 from further differential gene expression assessments. Conversely, high inter-sample correlations were confirmed among the remaining replicates in each group, validating the reliability of our experimental dataset. Figure 3B highlights the distribution of DEGs across treatments, revealing counts of 10, 350, and 546 unique DEGs for L vs. C, M vs. C, and H vs. C comparisons, respectively, with a commonality of two DEGs across all pairings.

Visual representation through a volcano plot (Figure 4A) underscores a substantial increase in DEGs upon comparing H vs. C, totaling 1036 DEGs, with 407 downregulated and 629 upregulated genes. Conversely, the M vs. C contrast saw a reduction in 417 DEGs (145 downregulated and 272 upregulated) (Figure 4B), while the L vs. C comparison yielded a mere 21 DEGs, characterized by 1 downregulated and 20 upregulated genes (Figure 4C). Intriguingly, these DEGs, both those that were downregulated and those that were upregulated, are implicated in a myriad of physiological processes, including carbohydrate and energy metabolism, lipid metabolism, amino acid metabolism, endocrine regulation, and immune responses (detailed in Table 1).

### 3.5. KEGG Enrichment

After completing the functional annotation of DEGs, we proceeded with a KEGG enrichment analysis to identify pathways harboring genes exhibiting statistically significant expression alterations. Our findings revealed that in the L vs. C comparison, no pathways achieved statistical significance (*p* > 0.05), as shown in Figure 5A. However, through an extensive cataloging effort, we found that 299 and 289 pathways were annotated for M vs. C and H vs. C comparisons, respectively, with 327 and 516 DEGs significantly enriched within these pathways.

As illustrated in Figure 5, the top 20 enriched metabolic pathways across each comparison were emphasized. Notably, the “Cysteine and Methionine Metabolism” pathway and the “Chemical Carcinogenesis-Reactive Oxygen Species” pathway were consistently enriched in both M vs. C and H vs. C comparisons. For M vs. C, the “Metabolic Pathways” stood out as the predominant enriched pathway. By contrast, within the H vs. C context, pathways affiliated with “Human Diseases” represented the most significantly enriched category. These results illuminate potential disruptions in biological processes and disease correlations linked to rising PFOS concentrations in crucian carp.

### 3.6. Validation of Significant DEGs by qRT–PCR

To reinforce the reliability of our RNA-seq findings, we performed qRT-PCR to validate the expression profiles of a select subset of 11 genes that displayed substantial differential expression, comprising five notably upregulated and six downregulated genes. The validation endeavor, as depicted in Figure 6, confirmed that the majority of these genes exhibited expression patterns consistent with those initially observed in the RNA-seq analysis. This agreement fortifies the precision and trustworthiness of the transcriptomic data garnered from our investigation.

## 4. Discussion

The ecotoxicological implications of perfluorooctane sulfonate (PFOS) have garnered significant scientific interest [41], with numerous studies focusing on its acute toxicity to aquatic organisms [42]. In vertebrate species, the 96 h 50% effective concentration (EC_50_) of the population has been quantified as 78.12 mg/L for zebrafish [43] while Giesy et al. [44] documented a more sensitive response in rainbow trout, with a 96 h LC_50_ of merely 7.8 mg/L. Our study contributes novel data by establishing a 96 h LC_50_ value of 23.17 mg/L for PFOS in crucian carp. This finding highlights the interspecies variability in susceptibility to PFOS toxicity and underscores the necessity for species-specific assessments in environmental risk evaluations, considering the unique biological characteristics of different fish taxa.

The liver, as the principal site for metabolism, lipid storage, and detoxification in fish, is particularly susceptible to PFOS accumulation and injury [45,46]. PFOS infiltrates primarily through ingestion or dermal contact and accumulates in the liver, where it undergoes partial metabolism by detoxification enzymes. However, its persistence and toxicity can overwhelm these defenses, causing hepatotoxicity [47]. Prior research revealed that chronic 0.5 μM PFOS exposure in zebrafish resulted in hepatocyte vacuolation and lipid droplet accumulation [24]. Our study echoes these findings, implicating oxidative stress-induced lipid peroxidation and disturbed glycolipid metabolism as contributors to vacuolation and triglyceride (TG) accumulation [48,49]. PFOS thus disrupts hepatic homeostasis, leading to structural abnormalities such as hepatocyte swelling and vacuolation.

While histological evidence clearly demonstrates liver damage by PFOS in crucian carp, the precise molecular mechanisms underlying this toxicity remain to be fully elucidated. Extensive research has shown that per- and polyfluoroalkyl substances (PFASs) possess hepatotoxic [50], immunotoxic [51], neurotoxic [52], reproductive toxic [53,54], and endocrine-disrupting properties [54]. Among these effects, alterations in the immune system appear to be one of the most sensitive health outcomes associated with PFASs exposure. To uncover the genetic underpinnings of PFOS-induced hepatotoxicity, we employed transcriptomic analyses on liver tissues exposed to PFOS using high-throughput RNA sequencing (RNA-Seq). This technique has emerged as a powerful tool for understanding the molecular impacts of pollutants on organisms [55], with the liver being a prime organ for such studies due to its central role in detoxification and metabolism [56,57].

Our study contributes to the limited transcriptomic knowledge regarding PFOS-induced liver injury in crucian carp, particularly under chronic exposure scenarios. Transcriptome profiling identified 1036 differentially expressed genes (DEGs) between control and high-PFOS-exposure groups, with 629 upregulated and 407 downregulated genes. Following KEGG enrichment analysis, these DEGs were classified into 289 pathways involved in biological processes, molecular functions, and cellular components. Notably, significant alterations were observed in pathways related to oxidative phosphorylation, neurodegeneration, thermogenesis, chemical carcinogenesis involving reactive oxygen species, and the citric acid cycle. Key pathways affected include those associated with the endocrine system, lipid metabolism, immune response, and amino acid metabolism. Specifically, changes were noted in genes involved in the TNF signaling pathway, the PPAR signaling pathway, the adipocytokine signaling pathway, fatty acid biosynthesis, and the fat digestion/absorption pathway (Figure 7). These findings align with previous observations that PFOS exposure induces significant expression changes in liver-related immune, endocrine, and metabolic pathways [58,59], corroborating our results which highlight lipid metabolism as a critical process impacted by PFOS.

Among the most notable changes are those in genes controlling hepatic lipogenesis, including members of the *PPAR* family, *AMPK*, and *SREBP* [60,61,62]. The altered expression of insulin receptor substrate genes (*IRS1/2*) suggests the potential development of hepatic insulin resistance [63], indicating an imbalance in glucose and lipid metabolism following PFOS exposure. These changes provide compelling evidence that PFOS may disrupt pathways of glucose synthesis, fatty acid oxidation, and lipid synthesis/storage in the hepatic parenchyma, potentially triggering nonalcoholic fatty liver disease (NAFLD). Furthermore, dysregulation of certain genes appears to exacerbate hepatocyte apoptosis, fibrosis, and inflammation, leading to liver injury. For instance, upregulation of pro-inflammatory cytokine genes such as interleukin-1 (*IL-1*), interleukin-6 (*IL-6*), and interferon-alpha (*INF-alpha*) supports the development of NAFLD [64,65]. Collectively, our data highlight the multifaceted effects of PFOS on liver physiology, suggesting that it can disrupt metabolic homeostasis and promote hepatic pathology through various interrelated mechanisms.

In light of the health implications for farmed fish from environmental pollution [66], our investigation into the chronic, low-dose effects of PFOS on crucian carp liver bridges significant gaps in ecological, toxicological, and biomedical knowledge. Our study provides novel insights into the molecular mechanisms of PFOS-induced liver damage in crucian carp, revealing the complex toxicity mechanisms and their profound implications for metabolic health. These findings underscore the urgent need to consider PFOS’s impact on aquatic life in conservation and ecosystem management strategies. Emphasizing the importance of further research, it is critical to address the variability among PFOS congeners, transgenerational impacts, and detoxification mechanisms. This will advance our understanding of PFOS’s ecological consequences and support the development of effective mitigation strategies to protect aquatic ecosystems. By addressing these aspects, future studies can provide a comprehensive framework for assessing and managing the risks associated with PFOS exposure, ultimately guiding efforts towards safeguarding both wild and farmed fish populations.

## 5. Conclusions

This study provides a comprehensive evaluation of PFOS toxicity in crucian carp using acute toxicity assays, histopathology, and transcriptomics. Key findings include a 96 h LC_50_ of 23.17 mg/L, highlighting significant PFOS toxicity. Histopathological analysis revealed severe liver damage, characterized by nucleolar dissolution, disorganized architecture, lipid accumulation, and hepatocellular vacuolation. Transcriptomic data show that PFOS alters gene expression linked to NAFLD progression, disrupting lipid metabolism, PPAR signaling, and hepatic fat processing. Exposure also impacted genes that are critical for endocrine and immune functions, exacerbating liver injury. These results deepen our understanding of the molecular mechanisms underlying PFOS-induced liver damage in crucian carp. Our study uniquely integrates lethal concentration data with molecular responses, offering new insights into the environmental impact of PFOS. By linking specific gene expression patterns to PFOS exposure, we contribute valuable information on pollutant effects on aquatic life. This research supports risk assessment strategies for persistent organic pollutants in freshwater ecosystems and informs conservation efforts to protect aquatic animal health. In summary, this work enhances our knowledge of PFOS’s impacts on aquatic organisms, providing essential data for assessing long-term risks and guiding policies aimed at ecological protection.

## Figures and Tables

**Figure 1 animals-15-00610-f001:**
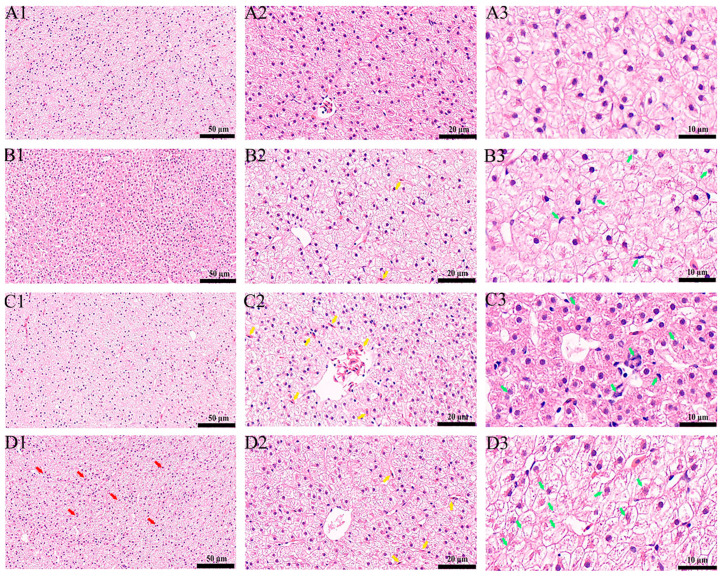
PFOS damages the liver of carp: (**A**): control; (**B**): 0.1 mg/L PFOS treatment group; (**C**): 0.5 mg/L PFOS treatment group. (**D**): 1.0 mg/L PFOS treatment group. (1, 2, 3): Microstructure of carp liver under 25×, 50× and 100× microscope. The red arrow: adipose tissue was significantly increased compared to the control group; the yellow arrow: compression deformation of hemoglobin compared to controls; the green arrow: deformation and lysis of nuclei compared to controls. (For interpretation of the references to color in this figure legend, the reader is referred to the Web version of this article).

**Figure 2 animals-15-00610-f002:**
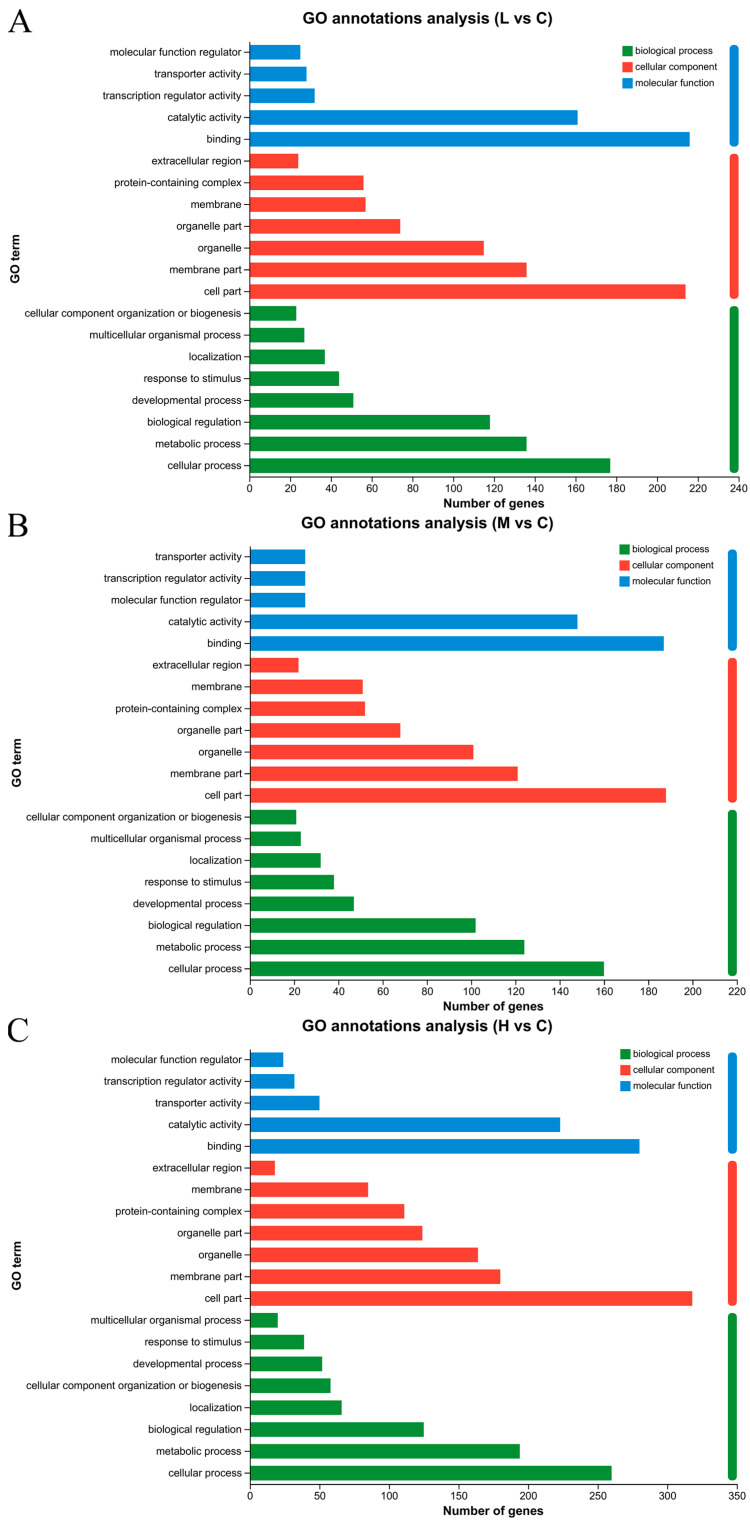
Annotation of the single genes based on the GO database: (**A**): 0.1 mg/L PFOS treatment group vs. control comparison (L vs. C); (**B**): 0.5 mg/L PFOS treatment group vs. control comparison (M vs. C); (**C**): 1.0 mg/L PFOS treatment group vs. control comparison (H vs. C).

**Figure 3 animals-15-00610-f003:**
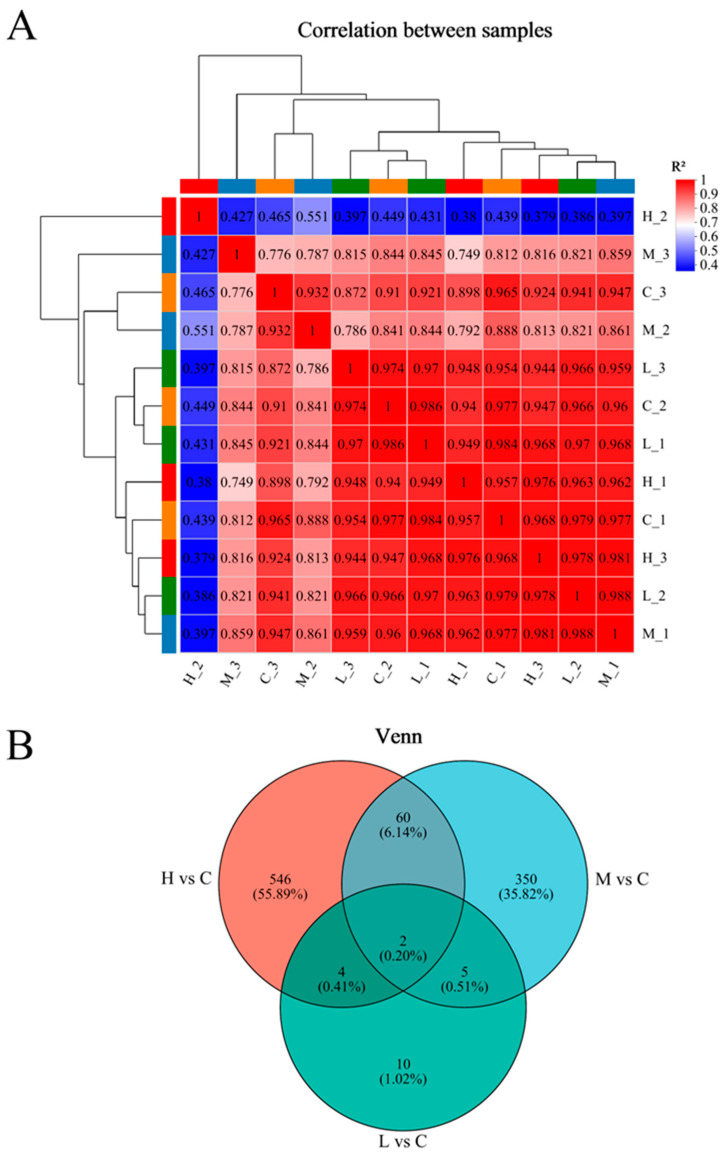
Heat map of inter-sample correlation (**A**) and Venn diagram analysis of DEGs (**B**). (**A**): High differential gene expression (red), low differential gene expression (blue). (**B**): Venn diagrams showing the DEGs between H vs. C (red), M vs. C (blue), and L vs. C (green).

**Figure 4 animals-15-00610-f004:**
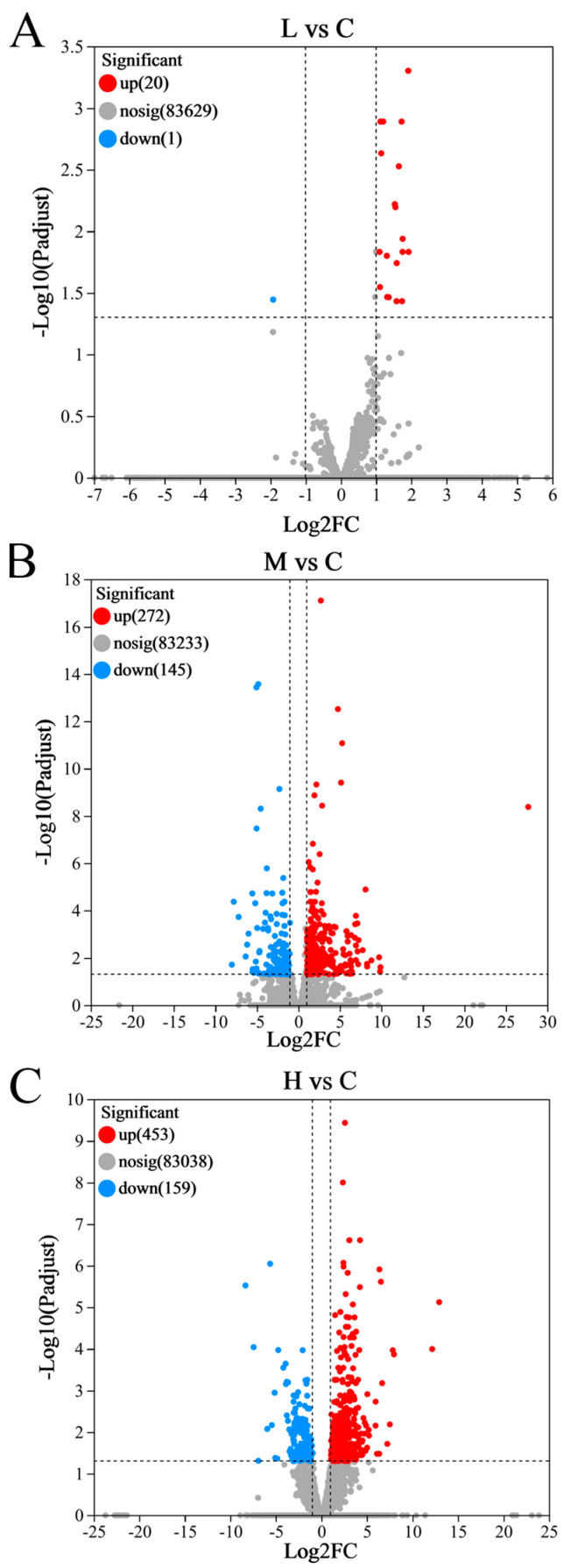
The volcano map shows the number of genes that have changed in the liver (FC ≥ 1.5 or ≤−1.5, *p* ≤ 0.05): (**A**): 0.1 mg/L PFOS treatment group vs. control comparison (L vs. C); (**B**): 0.5 mg/L PFOS treatment group vs. control comparison (M vs. C); (**C**): 1.0 mg/L PFOS treatment group vs. control comparison (H vs. C). Each dot represents one gene. Red and blue dots represent DEGs. Gray dots represent no differential expressed genes.

**Figure 5 animals-15-00610-f005:**
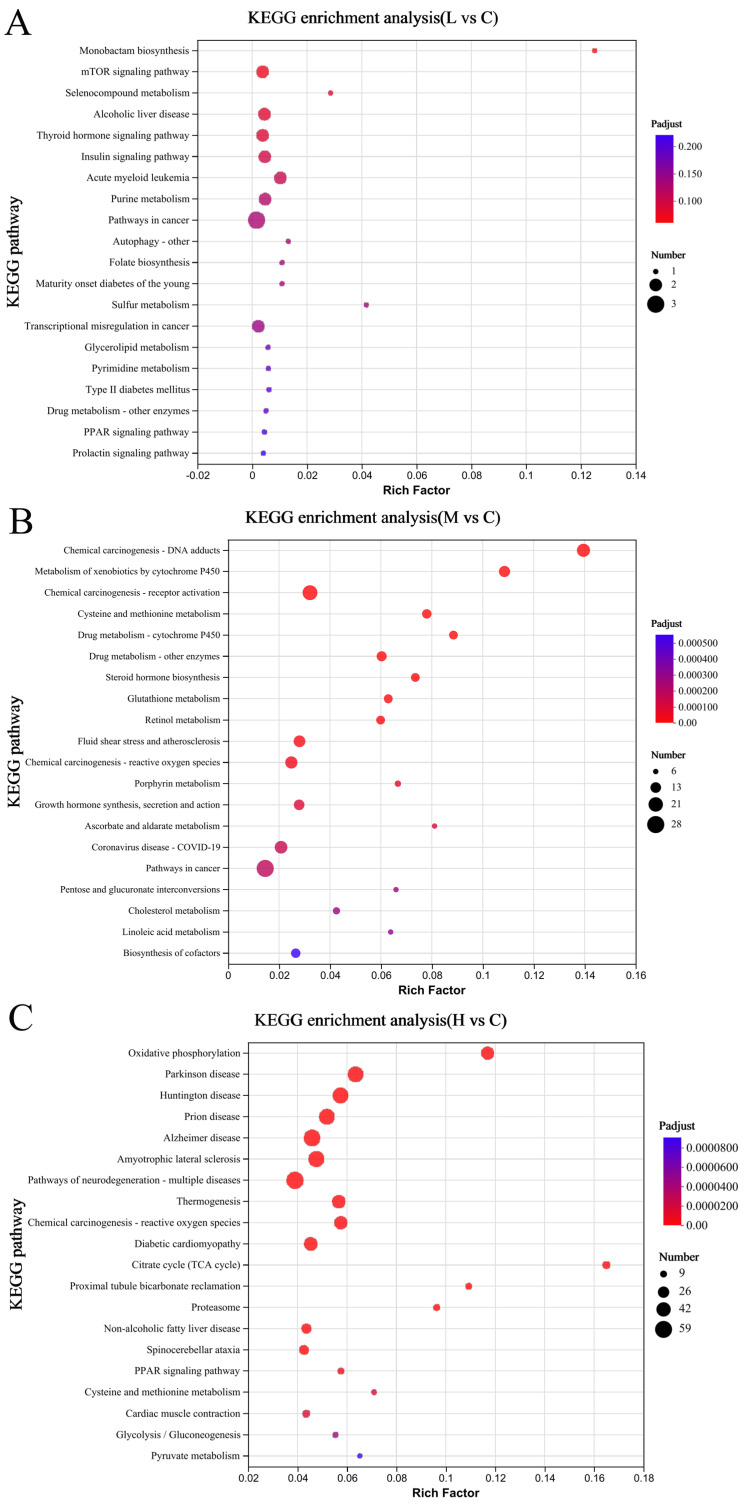
Comparative KEGG pathways analysis between (**A**) L vs. C, (**B**) M vs. C, and (**C**) H vs. C: (**A**): 0.1 mg/L PFOS treatment group vs. control comparison (L vs. C); (**B**): 0.5 mg/L PFOS treatment group vs. control comparison (M vs. C); (**C**): 1.0 mg/L PFOS treatment group vs. control comparison (H vs. C).

**Figure 6 animals-15-00610-f006:**
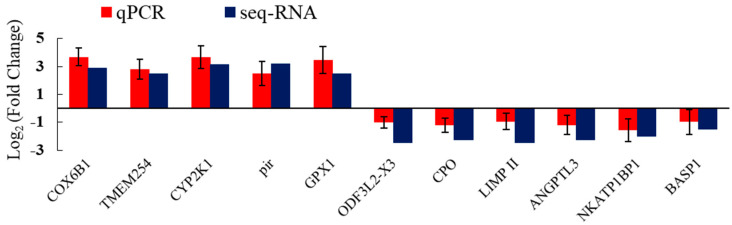
Comparison of gene expression data between RNA-Seq and qRT-PCR.

**Figure 7 animals-15-00610-f007:**
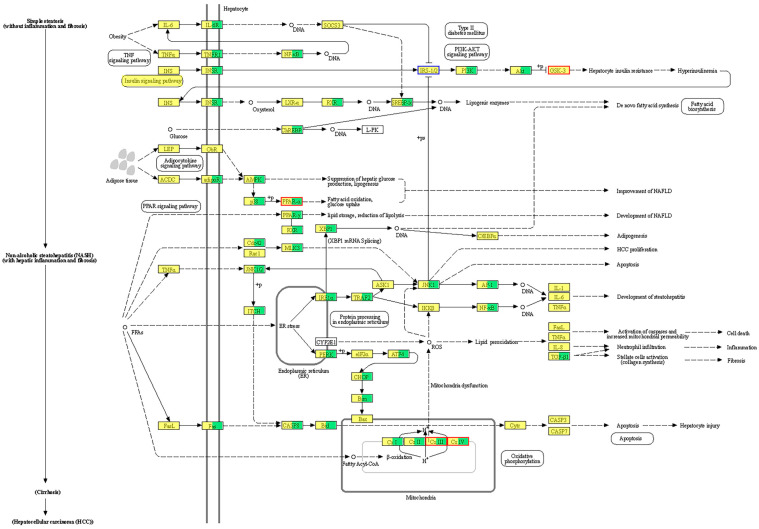
KEGG pathway of crucian liver treated with PFOS. Red boxes: significantly upregulated genes; blue boxes: significantly downregulated genes; yellow background color: known genes; green background color: new genes. The solid red boxes are the major signaling pathways and differential genes leading to non-alcoholic liver disease.

**Table 1 animals-15-00610-t001:** Some significantly differentially expressed genes.

Functions	Gene ID	Nr Description	log_2_FC	Up/Down
Carbohydrate metabolism and energy metabolism	LOC113040075	*ATP synthase F_1_ subunit gamma*	1.26	up
	LOC113106083	*NADH dehydrogenase [ubiquinone] Iron* *-sulfur protein 3*	1.20	up
	LOC113056135	*cytochrome c oxidase subunit 7A-Related protein*	1.75	up
	LOC113044453	*ATP synthase F _(0)_ complex subunit C_2_*	1.59	up
	LOC113078677	*ATP synthase subunit beta*	1.07	up
	LOC113077873	*Isocitrate dehydrogenase [NADP]*	1.97	up
	LOC113066956	*Cytochrome c oxidase subunit 5A*	1.53	up
	LOC113048234	*ATP synthase subunit gamma*	1.37	up
	LOC113108428	*ATP synthase subunit O*	1.16	up
Lipid metabolism	LOC113072791	*Glycerol-3-phosphate acyltransferase 3*	1.28	up
	LOC113067002	*Long-chain-fatty-acid--CoA ligase ACSBG1*	1.98	up
	LOC113049110	*Diacylglycerol o-acyltransferase*	1.20	up
	LOC113038637	*Carbonyl reductase [NADPH]*	2.23	up
	LOC113080231	*UDP-glucuronosyltransferase*	2.31	up
	LOC113041049	*Prostaglandin E synthase*	1.36	up
	LOC113092558	*Glycerol phosphocholine Phosphodiesterase*	1.71	up
Amino acid metabolism	LOC113043356	*Creatine kinase U-type*	2.57	up
	LOC113069635	*Branched-chain amino acid Transaminase*	2.52	up
	LOC113114441	*Alanine aminotransferase*	2.89	up
	LOC113051502	*Gamma-butyl betaine dioxygenase*	2.21	up
	LOC113040902	*Glutathione synthetase*	2.44	up
	LOC113057026	*Isoaspartylpeptidase/L-asparaginase*	2.70	up
	LOC113056857	*Aspartic transaminase*	2.36	up
Endocrine system	LOC113050872	*Cytochrome P450 27C1*	2.59	up
	LOC113056458	*Somatostatin receptor*	−3.47	down
	LOC113108922	*Insulin receptor substrate*	1.47	up
	LOC113110911	*Glutathion peroxidase*	2.51	up
	LOC113099998	*Thyroid hormone receptor alpha-A*	−1.62	down
	LOC113053072	*Sodium/potassium transport ATPase* *Subunit alpha-1*	1.28	up
	LOC113038699	*Sarcoplasmic/endoplasmic reticulum* *C* *alcium ATPase*	1.47	up
Heterotrophic bacteria biodegradation and metabolism	LOC113046050	*Uridine cytidine kinase 2-B*	2.19	up
	LOC113038637	*Carbonyl reductase [NADPH]*	2.23	up
	LOC113057403	*Glutathione S-transferase*	7.07	up
	LOC113080231	*UDP-glucuronidase*	2.31	up
Immune system	LOC113077942	*Cathepsin L1*	−2.73	down
	LOC113052117	*Thioredoxin*	1.98	up
	LOC113080830	*Interleukin-6 receptor subunit alpha*	0.55	up
	LOC113039569	*Interferon regulatory facto*	−1.67	down
	LOC113068555	*Heat shock protein HSP 90-beta*	1.69	up

## Data Availability

The raw data of RNA-seq have been deposited to the SRA database with the accession number PRJNA1193778.

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
