# Peer review of "Hepatotoxicity in Carp (Carassius auratus) Exposed to Perfluorooctane Sulfonate (PFOS): Integrative Histopathology and Transcriptomics Analysis"

_animals, 2025, doi:10.3390/ani15040610_

Round 1

Reviewer 1 Report

Comments and Suggestions for Authors

Please see my attested review report

Author Response

Dear Reviewer,
We appreciate the time and effort you have dedicated to reviewing our manuscript. Your constructive comments are invaluable to us and have significantly contributed to improving the quality of our work.
Please see the attachment for our detailed point-by-point response to each of your comments.

Reviewer 2 Report

Comments and Suggestions for Authors

This paper examines the exposure of crucian carp to perfluorooctane sulfonate (PFOS), encompassing both acute lethality and 21-day subchronic exposure assessments, with particular emphasis on hepatic effects. The findings reveal that PFOS exerts significant hepatotoxic impacts on crucian carp, disrupting critical pathways involved in lipid and energy metabolism, immunity, and endocrine regulation, all of which are essential processes in the development of nonalcoholic fatty liver disease (NAFLD). Overall, the article is well-structured and effectively presented. However, there are a few minor areas that warrant further refinement.

Comments:

1.The overall quality of the English expression of this manuscript needs to be improved. It is recommended that the manuscript undergo meticulous editing by an expert in technical English editing, with a focus on grammar, spelling, and sentence structure, to ensure that the objectives and results of the study are clearly conveyed to the reader.

2. Line 38 and Line 114: The term "96-hour lethal concentration 50% (LC50)" should be amended to "96-h 50% lethal concentration (LC50)". Please make the necessary revisions.

3. In Section 2.4 Histopathological examination, the fixation time for crucian carp liver tissues using 4% paraformaldehyde has not been specified. It is recommended to clarify this parameter. Furthermore, the description of the histopathological examination method can be made more concise and precise.

4. Please refine the conclusion section for greater conciseness, thereby more effectively highlighting the key findings and implications of the study.

Comments on the Quality of English Language

The overall quality of the English expression of this manuscript needs to be improved

Author Response

(The authors gave the same response as above.)

Reviewer 3 Report

Comments and Suggestions for Authors

I have reviewed the manuscript “Hepatotoxicity in carp (Carassius auratus) exposed to perfluorooctane sulfonate (PFOS): Integrative histopathology and transcriptomics analysis” with great interest. The authors present a thorough investigation into the hepatotoxic effects of PFOS on crucian carp, utilizing histopathological and transcriptomic analyses. While the study provides valuable insights into this specific toxicological issue, I have several recommendations for improvement, particularly regarding alignment with the journal’s scope, methodology, and presentation of results.

1. The manuscript primarily focuses on the toxicological effects of PFOS on the liver of crucian carp, which seems more specialized towards toxicology compared to the broader animal science research covered by the journal Animals. Given the journal’s audience and thematic coverage, it would be beneficial if the authors could bridge their findings with wider ecological or conservation biology issues, or explain the importance of this study in understanding fish health under specific environmental conditions.

2. In the introduction, it would be beneficial to mention the specific health concerns associated with PFOS exposure in humans and wildlife, to further emphasize the importance of the study.

3. please provide more details on the bioinformatics tools and software used for the RNA sequencing and DEG analysis.

4. In the section detailing transcriptomics analysis, Figure 3A reveals low intra-group sample correlation, especially within the high-dose group (H), where one sample (H_2) exhibits significantly different patterns from others. This low correlation may compromise the reliability and reproducibility of the results. The authors should revisit their experimental design, RNA extraction procedures, and sequencing quality controls to ensure consistent quality across all samples. Consider excluding outliers to improve dataset integrity if necessary. Increasing biological replicates could also strengthen the robustness of the conclusions.

Author Response

(The authors gave the same response as above.)

Reviewer 4 Report

Comments and Suggestions for Authors

The paper titled “Hepatotoxicity in carp (Carassius auratus) exposed to perfluorooctane sulfonate (PFOS): Integrative histopathology and transcriptomics analysis” focus on several approaches related to the characterization of the impact of long-term exposure to PFOS, using the Carassius auratus as a model system.

This work is very interesting because the impact of pollutant on organism is becoming an urgent issue.

The methodological approach is very accurate and the results are well schematized.

I suggest minor revisions.

L90. “A total of 200 juvenile crucian carps”. From a temporal point of view, what does juvenile correspond to?

L134: What is the medium in which the paraformaldehyde was solubilized? If the commercial substance was used, it is necessary to indicate it. Otherwise, more details on the preparation should be reported (temperatures, pH etc,,,).

Figure 1. I suggest placing the labels in such a way as to make the figure easy to read.

Figure 5-7. Compared to the other figures, the bold style is not shown.

Conclusions should be more fully argued. Authors should report what their results add to the existing literature and specify the value of their data, relatively to the macro-area of ​​ study.

Author Response

(The authors gave the same response as above.)

Round 2

Reviewer 3 Report

Comments and Suggestions for Authors

These responses have adequately addressed my concerns, leading me to suggest the acceptance of the manuscript.